# Performance of aging consumers in an e-commerce product choice task: The role of working memory and decision strategies

**Klara Rydzewska[1], Radosław Nielek[2]\*, Justyna Pawłowska[3], Adam Wierzbicki[3], Grzegorz Sedek[1]**

1 Interdisciplinary Center for Applied Cognitive Studies, SWPS University, Warszawa, Poland, 2 NASK–National Research Institute, Warsaw, Poland, 3 Polish-Japanese Academy of Information Technology, Warszawa, Poland

* nielek@pjwstk.edu.pl

**Data Availability Statement:** The dataset is publicly available in the OSF repository (https://osf.io/dx329/, DOI 10.17605/OSF.IO/DX329).

## Abstract

Over the past two decades, there has been a growing interest in research on aging and the decision-making behavior of older consumers. The subject of this article is multi-attribute decisions made using product comparison, a widely used functionality of many e-commerce stores. Studies on cognitive aging have established a negative relationship between age and accuracy in multi-attribute choice tasks; however, works in informatics (Human-Computer Interaction, UX research) have not accounted for how individual user differences affect the optimality of users' product comparison decisions. Our work attempts to bridge this gap between the disciplines of psychology of aging and informatics. We predicted and confirmed, in an online study simulating e-commerce shopping, the strong limitations of older adults in product comparison tasks. In subsequent modeling, other individual characteristics, such as visual working memory, were predicted and shown to be a reliable mediator of the relationship between age and decision accuracy. Popular product comparison tables are not sufficient for older consumers. Despite following UX guidelines for designing product comparison tables, overall correctness of consumer decisions in our study ranged from 90% to as little as 30%, depending on the difficulty of the task and the age of the consumer. These findings have important practical implications for UX design of e-commerce Websites.

## Introduction

E-commerce shopping adoption has recently strongly accelerated worldwide. Interestingly, this growth is fastest among older consumers. In many countries, online shopping has increased by threefold among older adults since the outbreak of the COVID-19 pandemic [1]. The pandemic is partially responsible for the accelerated e-commerce adoption by older adults, who's share in the population is increasing globally [2]. Over the past two decades, there has been a rising interest in research on the aging and decision-making behaviors of older consumers in shopping decisions, including e-commerce [3–5]. Current research in psychology,

**Funding:** This work was supported by grant 2018/29/B/HS6/02604 from the National Science Centre of Poland.

**Competing interests:** The authors have declared that no competing interests exist.

informatics and marketing is guided by the general idea that supporting older adults in making good decisions, especially regarding health and finances (including shopping decisions in e-commerce), is crucial for maintaining independent functioning and life satisfaction [6].

Virtually every consumer's purchase decision in a competitive market requires multi-attribute comparison of many products. The subject of this article is decisions made using product comparison, a widely used functionality of many e-commerce shops. The usage of product comparison tables has been shown to lead to improvements in shopping decisions, such as considering a larger share of nondominated products in the set of alternatives for purchase, as well as an increased probability of a nondominated product being selected for purchase [7]. Product A dominates another product B if A is not worse than B on any attribute, and A is better than B with respect to at least one attribute. Therefore, when compared to any other product, a non-dominated products is better with respect to at least one attribute. For example, consider three washing machines, A, B, and C. If A is non-dominated, than it has to be better than B on at least one attribute–for example, energy efficiency. At the same time, A has to be better than C on at least one attribute, for example, washing capacity. Note that, in previous work on product comparison tables, the question of making a choice among several non-dominated products has not been considered–yet this is a likely (and more challenging) scenario.

Product comparison is perceived as useful by all consumers, regardless of their personality traits and shopping characteristics [8]. However, experts in UX design recognize the threat of information overload in the usage of product comparison, leading to guidelines on the deployment of product comparison tables that require limiting the number of compared items and displayed attributes [9, 10]. Overall, research on product comparison has shown great opportunities for improving shopping decisions; however, at the same time, there exist threats in the use of this interface functionality.

Much research has been conducted on the effects of consumer age on UX [11–13], and methods proposed to improve UX of older users have focused on redesigning the visual interface to improve accessibility. Recent research has found that this is not sufficient; we need to look beyond the visual user interface, redesign user interactions, and support decision making to make a significant improvement of UX for older users [14]. Therefore, our long-term research goal is to redesign the product comparison functionality based on knowledge and research results in the field of the psychology of aging.

Systematically replicated research (see recent summary [15]) has established the existence of a negative relationship between age and accuracy in multi-attribute choice tasks. The authors used the battery of decision tasks named 'Decision-Making Competence'. The subset of decision tasks concerning multi-attribute choice was named 'Applying Decision Rules'. Older adults' decreased accuracy in applying relatively simple rules while solving multi-attribute choice tasks was strongly and negatively correlated with results of the fluid intelligence test using Raven's Standard Progressive Matrices [16] and with working memory measures [17–19]. In the above cited research that examined age differences in Applying Decision Rules, 10 specific multi-attribute choice problems were used. Each problem involved 5 DVD players that differed in terms of the following features: picture quality, sound quality, programming options, and reliability of brand. All feature values used the same scale and varied from 1 (worst value) to 5 (best value). Participants were asked to choose one DVD player by implementing several simple decision rules, such as selecting the DVD player with the highest average rating over attributes (called 'features').

Let us note several important differences in the format of the above described muti-attribute decision tasks (i.e., Applying Decision Rules) and standard e-commerce product comparisons. The product comparison tables in e-commerce shopping (see summary: [9]) use columns for the products or services and rows for the attributes. What is crucial is that

## A – most-difficult decision task

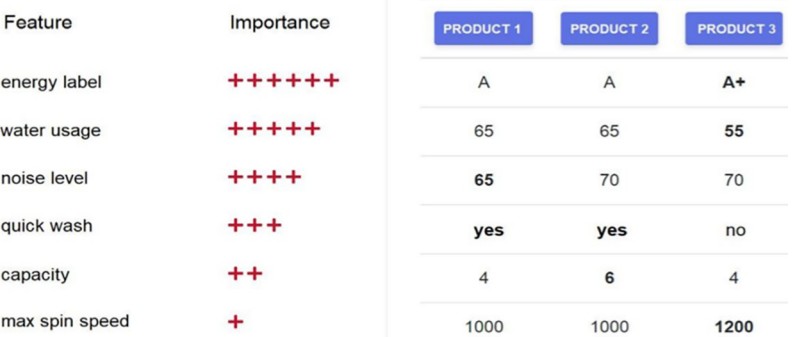

## B – simplest decision task

**Fig 1. Experiment screen showing a single multi-attribute choice task.** On the left are the washing machine features and their importance expressed by the number of plus signs. On the right are the three products and their features (the best feature values are in bold). The most-difficult version of decision task (**A**) required the use of the WADD strategy. Consequently, when using the WADD strategy, Product 2 should be selected (this would be the correct answer), when using the TALLY strategy, Product 1 should be selected (this would be the wrong answer), and when using the TTB strategy, Product 3 should be selected (this would be the wrong answer). For the simplest version of the decision task (**B**) when using any of the above decision strategies, Product 3 should be selected.

attributes have specific values on scales dependent on the meaning of the attribute. In our research (Fig 1), we applied the e-commerce format of comparison tables regarding washing machines: energy use is expressed in conventional labels, water usage in liters, noise in dB, etc. In this sense, our research faithfully reflects product comparison tables used in practice on e-commerce shopping platforms.

An additional aspect of product comparison underlined by Kate Moran in the report of Nielsen Norman Group [9] is that careful inspection of a small set of alternatives with their

original values helps consumers engage in *compensatory* decision making. Citing this report, 'People might accept a negative attribute as a tradeoff for a positive one. For example, a user researching a new laptop might be willing to consider a heavier computer if it has better battery life and computing power' [9]. Using another example from our research (Fig 1A), consumers searching for a suitable washing machine might be willing to select a product with a worse energy level if it has lower water usage and a lower noise level. Importantly, the tasks used in previous experiments (i.e., Applying Decision Rules) lacked decision rules that clearly require compensatory decision making.

We are familiar with only two psychological studies in the domain of aging and decision making that consider both compensatory (weighted additive) and noncompensatory strategies (TALLY, Take the Best) in the multi-attribute choice tasks [20, 21]. **Weighted Additive (WADD)** is an information-intensive strategy that involves multiplication of attribute values by attribute weights and addition of the weighted values for each alternative. The alternative with the largest sum should be selected. This strategy allows for compensation, where several attributes favoring one alternative can counterbalance another, even the most important attribute, favoring another alternative. A simpler strategy is **TALLY** [22], where attribute weights are ignored and the best attribute values are counted for each alternative. The alternative with the largest number of best attribute values is selected. **Take The Best (TTB)** is the simplest of the strategies considered here; it is a strategy based on the most important information (Fig 1A and 1B). This means making decisions by looking at the attribute with the highest weight and its respective values. If the most important attribute (one with the highest weight) is not discriminative, then TTB involves looking at the attribute with the second-highest weight and so on. Mata and associates [20] found that older adults (in comparison to younger adults) relied on simpler strategies (especially less on WADD). Additionally, older adults' increased reliance on simpler strategies was explained by an age-related decline in fluid intelligence (reasoning). These findings were replicated and extended in the research of Mata and associates [21]. The results confirmed that older adults had more difficulties in selecting the more cognitively demanding WADD strategy, often ignored attribute weight information and relied more on the simpler TALLY strategy. In both studies [20, 21], participants had only limited access to the features' information, and the format of the choice task was quite different from the solutions found in real online stores.

Note that while there exists related work in psychology on the multi-attribute choice problem, works in informatics (Human-Computer Interaction, UX research) have not considered how individual differences of users affect the optimality of their product comparison decisions. The reason for this knowledge gap is that in informatics experimental research, users were usually asked to make subjective decisions, and the experiments evaluated a user's subjective satisfaction [23, 24]. In studies that considered objective decision quality, the goal has typically been to evaluate the effectiveness of product comparison functionality by comparing a group of its users against a control group [25].

Our work attempts to bridge this gap between the disciplines of psychology and informatics. In the latter discipline, the answer to this research question is especially important. The underlying assumption that users of e-commerce systems make choices that are optimal or, at least, in some broad sense beneficial for themselves, is the basis of research on recommendation systems. Much research has been done with the goal of predicting purchase intent of consumers based on logs of their sessions on e-commerce platforms [26–29] and behavioral data, including mouse movements and keyboard clicks [30]. More advanced approaches also use eye-tracking data [31]. While researchers reported good results for predicting purchase intent, these predictions can also result in sub-optimal recommendations, if algorithms are trained on sub-optimal consumer choices. This phenomenon has been called *self-induced bias* [32] and is

a new type of data bias. State-of-the-art approaches toward removing biases that rely on increasing the frequency of data from the discriminated users would fail for this type of bias.

## The present study

Our study uses the format of product comparison matrices frequently used in e-commerce practice [9]. We do not instruct participants to use a specific strategy, but we presented a series of multi-attribute problems (in random order), some of them being rather simple: applying any rule—TTB, TALLY or WADD—allows us to identify the best choice (Fig 1B). Some of the multi-attribute problems are difficult and demand the application of the WADD rule (Fig 1A), and the rest are of medium difficulty (requiring the application of TALLY or WADD rules). In the first part of the task, participants received feedback on whether their choices were correct, and in the second part of the task, there was no feedback (see section Experimental Procedure for the details of the procedure of this study).

### Main predictions

We hypothesize (Hypothesis 1) that older adults demonstrate worse performance (than younger adults and middle-aged adults) even in applying simpler strategies such as TTB or TALLY (prediction based on age differences in applying decision rules, [15]. This prediction was confirmed in a pilot study [33]. In this study, there was no measure of working memory, and the difficulty levels of the choice tasks were not sufficiently differentiated. Nevertheless, this online study [33] used the same e-commerce format of selecting washing machines as the current study and showed a significant age-related decline in using the simplest TTB strategy.

It is also predicted (Hypothesis 2a) that age-related differences are most pronounced in choice tasks demanding the most cognitively costly compensatory strategy, such as WADD (prediction based on [20, 21]). The alternative view on the role of decision task difficulty in predicting age differences could be formulated based on Selective Engagement Theory (SET) formulated by Thomas Hess [34]. The SET explanation for selectivity effects in age-related cognitive performance is based on the assumption that the costs of cognitive engagement increase with age. According to this model, the cognitive effort required to achieve an objective level of task performance is disproportionately greater in old age. Extrapolating from the SET assumptions, older adults (in comparison to middle-aged and younger age participants) might be expected to exhibit longer processing time at each level of difficulty of decision tasks. Therefore, the alternative prediction (Hypothesis 2 B) based on the SET is that there will be main effects (and no interaction effect) of age and decision task difficulty on the performance in the multi-attribute decision task. That is, there will be stable age limitations at each level of difficulty of decision tasks (probably linear age trend), and more difficult decision tasks will result in worse decision task performance. Supplementing the above predictions based on the SET concerning processing time, main effects (and no interaction effect) of age and decision task on the processing time of the multi-attribute decision task might be expected (Hypothesis 3). That is, older participants (in comparison to middle-aged and younger adults) will exhibit longer processing times (independent of difficulty), and the more difficult the tasks, the longer the processing time will be. Based on strongly replicated findings relating age-related decline in using decision rules to age-related limitations in working memory [17–19], we predict (Hypothesis 4) that the age-related deficits in using the decision strategies in e-commerce product comparisons will be related to the age-related decline in working memory. The presence of feedback enables observation of learning and relies on a recent review of age differences in using feedback [35] and research on differentiating between more and less difficult

decision-making contexts [20, 21], which we predict (Hypothesis 5); therefore, learning from feedback will be reliable and similar in all age groups.

To our knowledge, the current study is the first comprehensive study that examines age-related competencies in using compensatory and noncompensatory strategies while using product comparison tables that faithfully resemble those used in practice for e-commerce shopping, as well as examining the role of individual differences in working memory.

## Subjective scales as predictors of performance of the multi-attribute choice tasks

In the context of the COVID-19 pandemic, we included our own measure of intellectual helplessness of contracting an infectious disease as an additional scale that may indicate the dysfunctional role of a high level of maladaptive emotions when making rational choices. The Scale of Helplessness of Contracting an Infectious Disease was inspired by the Intellectual Helplessness Scale initially developed for the education setting, where one tries to solve a task without understanding it for a prolonged period, and a lack of control ultimately leads to intellectual helplessness in the domain related to that task [36]. Similarly, futile efforts that accompany the uncontrollable situation of the pandemic (this online study was carried out before a COVID-19 vaccine was available) may lead to a state of cognitive exhaustion [35, 37] and overshadow rational choices. The pilot study [33] showed that the Scale of Helplessness of Contracting an Infectious Disease was indeed a significant predictor of worse performance in simple multi-attribute decision -making problems; therefore, we expect (Hypothesis 6) that the new Helplessness Scale might be a significant predictor of poor decision quality in the current study.

Numerical abilities are important for calculating and comparing attribute weights and values for several alternatives, as in the case of the multi-attribute choice tasks, so the numerical skills of participants were assessed with the Subjective Scale of Numeracy [38]. There is a large body of research regarding aging and numerical abilities, but this research focuses mainly on decisions of medical and financial importance, including medical insurance plans or prescription drug programs [39, 40], judgments about probabilities of developing specific illnesses, or health risks associated with specific medical procedures [41]. Little is known about the difference across age groups in terms of the effect of numeracy on other decisions that are not medically or financially related [35]. However, the pilot study [33] showed that subjective numeracy is a significant predictor of simple multi-attribute decision-making; therefore, we expect (Hypothesis 7) that it might still be a significant predictor of decision-making (not only for simple but also for more difficult tasks) in the current study.

Moreover, participants completed the Need for Cognitive Closure Scale [42], designed to measure epistemic motivation at the interindividual level. Precisely, it measures the degree of the desire for a definite answer to a question or a problem, as opposed to confusion, ambiguity, or uncertainty. The general results from numerous our studies are that the need for cognitive closure increases with age, especially for two of the subscales: preference for order and structure and preference for predictability [36, 43–45]. Despite such a general tendency toward an increase in the need for cognitive closure with age, there are individual differences in its level, with some older adults being relatively low or high in the need for cognitive closure. Interestingly, older adults with a relatively low need for cognitive closure are more flexible in computerized sequential decision tasks [45]. That is why we predict (Hypothesis 8) that older adults with a relatively low need for cognitive closure (in contrast to older adults with a relatively high need for cognitive closure) will show better performance in multi-attribute product choice tasks.

## Method

### Participants

The experiment was conducted online on the Ariadna panel (https://panelariadna.com/). Ariadna is the largest independent nationwide online research panel in Poland. The study was approved by the SWPS University Ethics Committee (opinions no. 14/2020). The data was collected between 9th and 18th May 2021. Each participant provided informed consent by clicking on the clearly marked button. Participants in the study were remunerated using points on the Ariadna platform that can be exchanged for rewards (products) and were motivated to perform well by receiving extra points for an above-average (60%) level of performance. The preliminary sample of participants recruited in our online study consisted of 170 participants. Twenty-one participants were excluded from the analyses (10 younger adults, 12 middle-aged adults, and 9 older adults) because they had very low level performance in the simplest decision task (accuracy below 70%) and their decision times were below the general mean. The removal of these participants was carried out using principles of spammer detection and removal from crowdsourcing research [46, 47].

The final sample consisted of one hundred forty-nine participants, including 50 younger adults (25 women; age range 19–30; mean age = 25.72, SD = 2.86; mean years of education = 15.16, SD = 2.60), 45 middle-aged adults (22 women; age range 42–53; mean age = 47.82, SD = 3.37; mean years of education = 16.18, SD = 3.31) and 54 older adults (28 women; age range 65–76; mean age = 69.37, SD = 3.03; mean years of education = 15.43, SD = 3.04). Participation in the study took between 1 and 2 hours, and the participants were remunerated for their participation. The study was approved by the Ethics Committee of the SWPS University in Warsaw.

### Products, attribute values, preferences and product choice tasks

Fig 1 shows the main screen of a single experimental task. Participants were instructed to select the best washing machine out of 3 available products, taking into account all the features' values and their importance, expressed as the number of plus signs associated with each feature. The importance order, that is, the order of features from the most important (6 plus signs) to the least important (1 plus sign), was determined using a survey collected during a separate study. We chose to use washing machines in our experiment because they are well-known everyday appliances. Additionally, they have many attributes that are well understood by most consumers. Importantly, the features included in the task were realistic for currently available washing machines. We chose not to include price in the list of the features because we assumed that e-commerce clients will typically choose products of similar price for comparison.

Note that the design of our experiment follows the guidelines of designing product comparison tables for good UX–see the report Nielsen Norman Group [9]. We limit the number of considered attributes to six, and the number of compared products to three.

As discussed in the Introduction, previous work in psychology that studied age effects on multi-attribute choice [15] has used simplified forms of multi-attribute choice problems. In practical e-commerce systems, product comparison tables use real values for attributes, and each attribute has its own value scale. In this study, we wanted to investigate whether the findings of psychological research on multi-attribute choice can be replicated for e-commerce product choice. To achieve this goal, we have chosen to closely reproduce the properties of real product choice problems, while keeping the basic design of multi-attribute choice experiments to allow the quality of participants' decisions to be evaluated. In the first step, we decided to choose attribute importance ratings based on a survey of experiment participants. To avoid

affecting the current experiments, we chose to carry out the survey in a pilot study. Participants in this separate study (120 women and 72 men, age range 52–75) were instructed to rate the importance of selected features (attributes) of washing machines (energy level, water usage, noise level, quick wash function, capacity and maximum spin speed). The importance rank of attributes in the current experiment was determined by average importance ratings given by participants in the pilot study.

The second step of preparing the experimental tasks consisted of preparing the values of product attributes. The set of 64 tasks, each consisting of 3 products compared by the participants, was prepared to reflect real-life choices. To ensure that the washing machines' parameters are close to those observed on the market, we web-scraped an assortment of popular household appliances available on e-commerce stores. For 6 key attributes of washing machines, we collected data on the statistical distribution of values in the population to determine what can be considered the norm for high-end and low-end products and what is the typical variation among the values. The details of the features are described in (S4. Description of the Features in S1 File).

Additionally, we introduced two parameters for the product comparisons: quality (Q) and delta (D). Quality differentiate two versions:: a selection of high-end products and a selection of low-end products. The delta parameters specified the difference between better and worse parameter values within the same task. The details of parameters are described in (S5. Description of the Quality and Delta Parameters in S1 File).

A final step in the preparation of experimental tasks was the control of task difficulty. A product choice task is simple if applying any rule–TTB, TALLY or WADD–allows the best choice to be identified (Fig 1B). On the other hand, product choices with medium difficulty demand the application of TALLY or WADD rules, while product choices with high difficulty require the application of the WADD rule to obtain a correct result (Fig 1A). Fifty percent of the product comparison tasks in our experiment were of low difficulty, 25% were of medium difficulty, and 25% were of high difficulty.

## Materials

**Measure of Visual Working Memory (VWM).**   As the measure of VWM, we applied the visual pattern span task that was successfully applied in a large internet study with participants across the adult life span [48]. A rectangular matrix pattern with white and blue squares was presented for 2 seconds and then replaced with a blank matrix; the task was to correctly select the squares that were previously blue. The patterns started with a matrix of 3 squares x 2 squares (4 blue squares), and the matrices increased up to a maximum of 5 squares x 5 squares (12 blue squares), with two patterns shown at each level (matrix size). The test stopped when participants failed to correctly recall blue squares on two consecutive trials of a given matrix size. Performance was scored as the number of patterns correctly recalled.

**Measures of Subjective Numeracy, need for cognitive closure, and Helplessness of Contracting an Infectious Disease.**   Participants completed the Scale of Subjective Numeracy [38]. This scale consists of seven questions measuring the subjective view of cognitive abilities in relation to fractions and percentages and preferences for displaying numeric information. The reliability of the Subjective Numeracy Scale was high: Cronbach's alpha = .87.

Next, participants completed the Need for Cognitive Closure Scale (42), which includes 12 statements that measure epistemic motivation on the interindividual level. More specifically, the scale measures a desire for a definite answer to a question, as opposed to ambiguity, uncertainty, or confusion, regardless of the answer's quality. The Cronbach's alpha of this scale was .74.

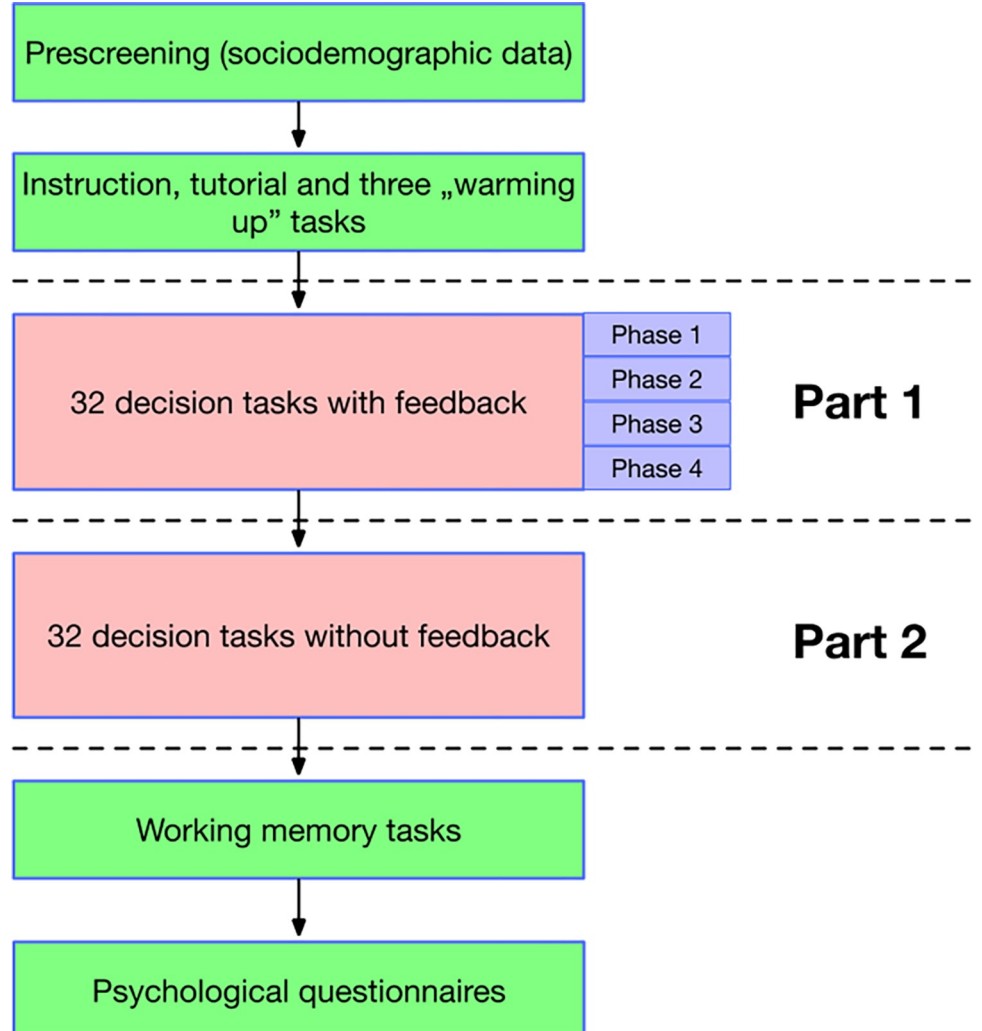

**Fig 2. Experimental study procedure.** The experiment was divided into two parts: Part 1 with feedback and Part 2 without feedback. The 32 tasks of Part 1 were further divided into four phases of 8 tasks each to study the effect of feedback on the learning process in more detail.

Finally, participants completed the Scale of Helplessness of Contracting an Infectious Disease. This scale is composed of ten items answered on a 5-point Likert scale ranging from 1 = "never" to 5 = "always". An exemplary item is "I feel helpless in the face of the possibility of contracting an infectious disease". The reliability of this scale was impressively high: Cronbach's alpha = .93.

## Experimental procedure

The experiment was conducted online on the Ariadna panel (https://panelariadna.com/). Participants were redirected from Ariadna's portal to the experiment's website. Participants completed the experiment at home using their own computers and preferred internet browser. At the beginning of the study (see Fig 2 for a graphical presentation of the experimental procedure), participants reported their sociodemographic data (age, sex, years of formal education). The instructions were then shown to participants, followed by four training trials for the

multi-attribute choice tasks. In each task, participants were instructed to select the best products based on the importance of a given attribute (number of pluses). Then, the participants were informed that the training had ended and directed to the main part of the experiment, in which they had to solve 64 multi-attribute product choice tasks. We did not instruct participants to use a specific strategy, but we presented a series of 64 multi-attribute problems (in random order, the same for all participants), 50% of them being rather simple: applying any rule–TTB, TALLY or WADD–allows us to identify the best choice (Fig 1B). Twenty-five percent of the multi-attribute problems were difficult and required the application of the WADD rule (Fig 1A), and an additional 25% were of medium difficulty (demanding the application of TALLY or WADD rules). As described in the Method section, half of the presented products had a high Q parameter (quality), and half of the products had worse values. Analogically, half of the presented products had large values of the D parameter (large differences), and half had small D values (small differences).

In the first part of the task (32 problems, with the percentage of problems of varying difficulty presented above), participants received feedback on whether their choices were correct, and in the second part of the task, there was no feedback. Participants then assessed the subjective importance of washing machine attributes, followed by a visual working memory test. Finally, participants completed the questionnaires listed in the Materials section.

## Results

First, we will describe the main results: the effects of age and task parameters on the accuracy of decisions and decision time. Next, we will provide an overview of the correlation matrix of the main variables. In the case of interactions which did not involve the age variable, the results and figures are reported in S1 File. Finally, we will demonstrate the role of cognitive, motivational, and emotional variables as potential mediators, moderators, and covariates of the relationship between age and multi-attribute decision-making task performance.

### Proportion of correct decisions

**Part I of the decision task–the role of feedback.** The key measure of performance in the multi-attribute product choice task is the proportion of correct decisions. In Part I, participants received feedback concerning the accuracy of their choices in the task (appropriate to the strategy demanded by the given decision task, i.e., WADD, TALLY or TTB). The 32 trials were grouped into four phases to observe the potential role of feedback in improving performance within this first part of the decision task.

A 3 x 3 x 4 (Age [younger adults, middle-aged adults, older adults] x Decision Difficulty [simple, moderately difficult, difficult; within-subject variable] x Phase of the task [first, second, third, fourth; within-subject variable]) mixed ANOVA on the proportion of correct decisions yielded three main effects and an interaction effect. There was a main effect of age, $F(1, 146) = 8.82$, $MSE = .28$, $p < .001$, $\eta_p^2 = .108$. The performance of younger adults ($M = .71$) was significantly higher than that of both the middle-aged ($M = .62$, $p = .03$, for this and next comparisons, Sidak post-hoc tests were applied) and older adults ($M = .58$, $p < .001$), and there were no significant differences between the middle-aged and older adult groups. There was also a main and strong effect of decision difficulty, $F(2, 292) = 138.00$, $MSE = .18$, $p < .001$, $\eta_p^2 = .486$, with significant differences ($p < .001$,) between all decision tasks; the more difficult the task was, the worse the performance. Finally, there was also a main effect of Phase, $F(3, 438) = 47.02$, $MSE = .07$, $p < .001$, $\eta_p^2 = .244$. There was a significant increase in performance in phase 2 in comparison to phase 1 ($p < .001$), with a lack of further significant progress in

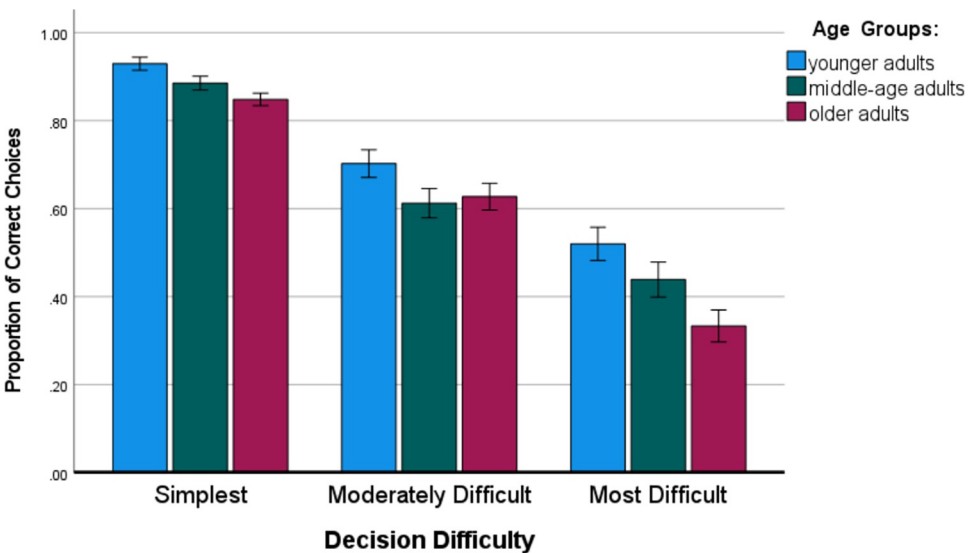

**Fig 3. Proportion of correct choices in the multi-attribute decision task as a function of age group and decision difficulty.** Error bars represent standard errors.

phase 3 and phase 4. The last two main effects were qualified by the significant Decision Difficulty x Phase interaction (see S1. Decision Difficulty x Phase interaction in S1 File).

**Comparison of part I and part II of the decision task.** In Part I of the task, participants received feedback concerning the accuracy of their decision making in 32 trials of decision tasks of varied difficulty. In Part II, participants also performed 32 decision tasks of varied difficulty but without feedback. We examined the potential transfer effect (whether the performance in Part 2 is higher than that in Part 1 because of learning from the feedback), as well as the stability of age and difficulty differences across both parts of the decision task.

A 3 x 3 x 2 (Age [younger adults, middle-aged adults, older adults] x Decision Difficulty [simple, moderately difficult, difficult; within-subject variable] x Part of Decision Task [I, II; within-subject variable]) mixed ANOVA on the proportion of correct decisions yielded three main effects and an interaction effect. There was again a main effect of age, $F_{(2, 146)} = 8.06$, MSE = .13, $p < .001$, $\eta_p^2 = .099$. The performance of younger adults was marginally or significantly higher than that of both the middle-aged adults ($p = .054$) and the older adults ($p < .001$). Additionally, the decreased linear trend of age on decision performance was highly significant, $t_{(146)} = 15.89$, $p < .001$.

There was also again a main and strong effect of decision difficulty, $F_{(2, 292)} = 228.91$, MSE = .07, $p < .001$, $\eta_p^2 = .611$, with significant differences ($p < .001$) between all decision tasks; the more difficult the task was, the worse the performance. Fig 3 presents the pattern of main effects of age and decision task difficulty.

Additionally, linear trends of age on decision tasks of various difficulties showed strong decreased linear trends for simplest decision tasks, $t_{(146)} = 3.98$, $p < .001$ and most-difficult tasks, $t_{(146)} = 3.56$, $p < .001$. However, such a linear trend was only marginally significant for moderately difficult tasks, $t_{(146)} = 1.69$, $p < .09$.

Finally, there was also a main effect of the part of the decision task, $F_{(1, 146)} = 15.03$, MSE = .02, $p < .001$, $\eta_p^2 = .092$. There was a significant increase in performance in Part II of the decision task in comparison to that in Part I. The last two main effects were qualified by the significant Decision Difficulty x Part of Decision Task (see S2. Decision Difficulty x Part of Decision Task interaction in S1 File).

Moreover, we examined the role of the quality (Q) parameter (whether the compared washing machines were of high-end or low-end quality and their price range) and the role of the delta parameter (whether the compared washing machines' features had high or low variability). The results of a mixed ANOVA 3 x 2 x 2 (Age x Quality x Delta) across all trials of multi-attribute choice tasks did not yield any significant results.

## The role of decision times

The previous analyses showed strong main effects of age and task difficulty on the accuracy of the multi-attribute choice tasks. Next, analyses aimed to demonstrate whether participants in different age groups noticed differences in difficulty of the decision tasks and devoted more time to choice tasks demanding more sophisticated strategies or if the decision time was similar, indicating that the time-consuming guessing strategy was used in more complex tasks. Due to a small number of extremely high times for some decisions in the online study, we used medians instead of means as measures of central tendency. Additionally, the distributions of decision time indices were carefully examined to potentially exclude extreme values (there were no such cases).

A 3 x 3 x 2 (Age [younger adults, middle-aged adults, older adults] x Decision Difficulty [simple, moderately difficult, difficult; within-subject variable] x Part of Decision Task [I, II; within-subject variable]) mixed ANOVA on the time of decisions yielded three main effects. There was a main and strong effect of age, $F (2, 146) = 29.75$, MSE = 161.36, $p < .001$, $\eta_p^2 = .290$. The decision times of older adults were significantly higher than those of both the middle-aged group and the younger adult group ($p < .001$), and no significant differences between younger adults and middle-aged groups were observed. Additionally, the increased linear trend of age on decision time was highly significant, $t (146) = 6.93$, $p < .001$.

There was also a main and strong effect of decision difficulty, $F (2, 292) = 61.66$, MSE = 24.4, $p < .001$, $\eta_p^2 = .297$, with significant differences ($p < .005$) between all decision tasks–the more difficult the task was, the longer the decision time. Fig 4 presents the pattern of main effects of age and decision task difficulty for time taken to make multi-attribute choices.

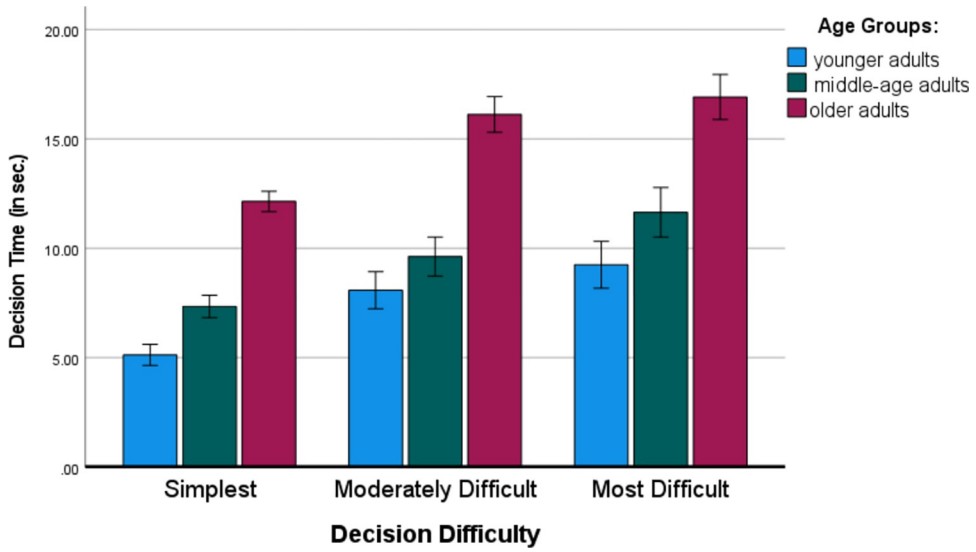

**Fig 4. Means of decision times in multi-attribute decision task as a function of parts of the choice task and decision difficulty.** Error bars represent standard errors.

Finally, there was also a main and strong effect of part of the decision task, $F(1, 146) = 41.70$, MSE = 23.39, $p < .001$, $\eta_p^2 = .222$. There was a significant decrease in decision time in Part II in comparison to Part I (M = 11.74 sec. vs. M = 9.65 sec.).

## The effect of age on decision accuracy: The mediating role of visual working memory and moderating role of helplessness of contracting an infectious disease

In the subsequent analyses, we applied measures of accuracy of multi-attribute choices and measures of decision times as appropriate means or medians of the simplest, moderately difficult and most-difficult decision tasks. The correlation tables between accuracy of the decisions, age, decision time, visual working memory and questionnaire measures (see Table 1 for correlation for the entire sample and S1. Decision Difficulty x Phase interaction for correlations within three age groups in S1 File) indicate that only visual working memory correlated positively and significantly with decision accuracy for the whole sample, and these correlations were also significant within each age group. Because visual working memory also correlated strongly with age (in years), it was predicted that in subsequent modeling, it would be a reliable mediator of the relationship between age and decision accuracy. These correlations also indicated that Helplessness (of Contracting an Infectious Disease) correlated negatively and significantly with decision accuracy not only for the whole sample (Table 1) but also within two age groups (younger and middle-aged adults, S1. Decision Difficulty x Phase interaction in S1 File). Because intellectual Helplessness also correlated negatively with age (in years), it was predicted that in subsequent modeling, it would be a reliable moderator of the relationship between age and decision accuracy. The Subjective Numeracy measure was examined as potential covariate of the above mediating or moderating analyses.

To examine the reliability of the expected mediation and moderation effects, we applied Process software [49], and conceptual model 5 indicated that aging was significantly related (t = 7.22, p < .001) to visual working memory (VWM) and that VWM was a reliable mediator (t = 4.29, p < .001) of the relationship between aging and the proportion of correct answers in the decision task. Parallelly, this mediating model was enriched by the conditional age x intellectual helplessness moderation (t = 3.00, p < .01) of the direct effect of age on decision making (Fig 5).

We applied 10000 bootstrap samples to estimate percentile bootstrap confidence intervals. To integrate these findings, conceptual model 5 is supplemented by one covariate. Subjective

**Table 1. Mean, standard deviation, and correlation table of dependent and independent variables.**

| Variable | Mean | SD | 1 | 2 | 3 | 4 | 5 | 6 | 7 |
|---|---|---|---|---|---|---|---|---|---|
| 1. Age | 48.21 | 18.58 | - | | | | | | |
| 2. Acc_Dec | .65 | .15 | -.35** | - | | | | | |
| 3. Tm_Dec | 10.88 | 6.14 | .50** | .04 | - | | | | |
| 4. VWM | 3.02 | 1.27 | -.53** | .50** | -.30** | - | | | |
| 5. Helplessness | 2.52 | .87 | .39** | -.30** | .10 | -.33** | - | | |
| 6. S Numeracy | 3.88 | .95 | -.14 | -.30** | -.09 | .25** | -.12 | - | |
| 7. NFC_4subs | 3.90 | .49 | .22** | -.17* | .04 | -.22** | .16* | -.01 | - |

*Note*: Acc_Dec = Accuracy of Decision Task; Tm_Dec = Time of Decision Task; VWM = Visual Working Memory Task; Helplessness = Scale of Helplessness of Contracting an Infectious Disease; S Numeracy = Subjective Numeracy Scale; NFC_4subs = 4 subscales of the Need for Cognitive Closure Short Scale

*p < .05

**p < .01.

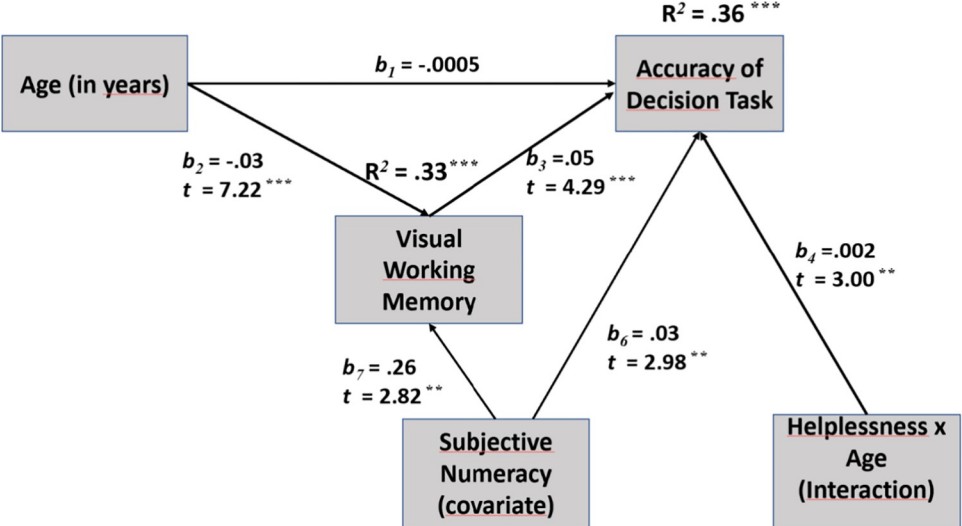

**Fig 5. Mediational model with moderation effect, which indicates the interplay between the age-related limitation and the role of additional variables (especially VWM, Helplessness) in explaining the relationship between aging and performance in the multi-attribute choice task.**

numeracy was a significant predictor of both VWM (t = 2.82, p < .01) and decision accuracy (t = 2.98, p < .01). Interestingly, in this integrated model, the variance in decision accuracy explained by the mediator, moderator and covariance variables was relatively high ($R^2$ = .36, p < .001), and there was no significant direct effect of age on decision accuracy.

## Discussion

### Summary of the main findings in relation to the hypotheses

In our study, we examined the age differences in performance on the product choice task. The results confirmed most of the hypotheses and supported some alternative predictions. There were reliable differences between age groups in Part I of the study (the best performance of younger adults, supporting Hypothesis 1), as well as difficulty of decision tasks (more difficult tasks lead to worse performance) with no interaction between age and difficulty of the tasks (no support for Hypothesis 2a). There was a significant improvement in the performance of the simplest tasks across all phases of Part I of this study (with constant feedback), while in the more-difficult tasks, there was improvement in only phase 2 in comparison to phase 1, with a lack of further progress in the next three consecutive phases. The pattern of this improvement was similar for all age groups (no significant interaction with age, supporting Hypothesis 4). Briefly summarizing the main findings for the two parts of the decision tasks, there were reliable and robust differences between age groups (the best performance of younger adults, linear decrease of performance with age) and between the difficulty of decision tasks (more difficult tasks lead to worse performance). There was also a significant improvement in performance on the simplest and moderately difficult tasks in the second part of the study (without feedback), showing a transfer effect of learning from the first part of the study (with feedback). The pattern of this improvement was similar for all age groups (no significant interaction with age). The established age-related limitations in the performance of multi-attribute product choice tasks, especially in the older adult group, partly confirm previous findings from studies that usually compared only younger and older adults. More specifically, the strong limitations of older adults (in comparison to younger adults) in the most-difficult multi-attribute choice

tasks are in line with previous laboratory experiments [20, 21]. A novel finding of our online study that imitated practical e-commerce shopping is the strong decreasing linear trends in both the simplest and most-difficult multi-attribute choice tasks across three age groups of adult participants, including the middle-aged group. We have also shown that the distribution of product attribute values in tasks (as controlled by the Q and D parameters) does not have a significant effect on the results, especially compared with task difficulty (recall that half the tasks involved a comparison of high-end and half of low-end products).

To briefly summarize the main findings for decision time when making multi-attribute choices, there were reliable and robust differences between age groups (linear increase in decision time across age groups), which confirms Hypotheses 2A and 3 based on the extrapolation of Selective Engagement Theory [34, 50]. To our knowledge, the findings that participants in all age groups noticed the increased complexity of the moderately and most-difficult tasks and spent more time in making decisions in those trials requiring compensatory strategies is novel and important. There are some paradoxes in this pattern of results: participants spent significantly more time on the more-difficult tasks, but their performance was significantly worse in those more-complex choice tasks. It seems that users of product comparison tables lack the appropriate cognitive skills to implement compensatory strategies, although they are ready to deliberate longer on more-complex choice tasks. This suggests that users might be receptive (and ready to invest more time in decision-making) to decision support systems or agents helping them to achieve better performance in more demanding multi-attribute choice tasks.

The correlation tables between the accuracy of the decisions, age, decision time, visual working memory and questionnaire measures (see Table 1 for correlation for the entire sample) supported the predictions (Hypotheses 4, 6, and 7) that visual working memory, Scale of Helplessness of Contracting and Infectious Disease, and Subjective Scale of Numeracy are significant predictors of accuracy in multi-attribute choice. However, we did not observe any role of the need for cognitive closure (lack of support for Hypothesis 8).

Only visual working memory correlated positively and significantly with decision accuracy, not only for the whole sample, but also for each age group (see S3. Correlation of Independent Variables Separately for Three Age Groups in S1 File). Since visual working memory was also strongly correlated with age (in years), it was predicted and demonstrated in subsequent modeling to be a reliable mediator of the relationship between age and decision accuracy.

Lindenberger and Pötter [51] underlined the limitations of regression mediating models using extreme age groups in providing a reliable account of variance explained because of existing intercorrelations between predictor variables. The primary problem is that correlations between two variables that are each correlated with age may result in a spurious relationship between the two variables that does not reflect a true relationship. However, there are some counter arguments against the possibility of such spurious relationships in our analyses. First, in our analyses, we rely on three age groups (including the middle-aged group) and not on extreme age groups (i.e., older adults vs. younger adults). Second, the zero-order correlations between visual working memory choice performance were significant for each age group. To our knowledge, our findings are the first to demonstrate in a methodologically correct manner the role of visual working memory as a mediator of the relationship between age (in three age groups) and decision accuracy in an e-commerce product choice task.

These correlations also indicated that Helplessness (of Contracting an Infectious Disease) correlated negatively and significantly with decision accuracy not only for the whole sample (Table 1) but also within two age groups (younger and middle-aged adults, see S1. Decision Difficulty x Phase interaction in S1 File). Therefore, it was predicted and found in subsequent modeling that it was a reliable moderator of the relationship between age and decision accuracy. The other questionnaire measure of Subjective Numeracy were examined as potential

covariates of the above mediating or moderating analyses. The examination of Subjective Numeracy as another potential moderator of the relationship between age and decision accuracy did not yield the significant results.

The mediational model with moderation effect and covariate nicely integrates the correlational analyses and indicates the interplay between the age-related limitation and the role of additional variables (especially VWM, Helplessness) in explaining the relationship between aging and performance in the multi-attribute choice task. However, we are aware that this mediational model, although original, is still preliminary, and explaining the psychological mechanisms behind the role of VWM as a mediator and intellectual helplessness as a moderator demands further experimental work (e.g., experimental manipulation of the intensity of a mediator and moderator, see [52]).

### Implications for the integration of psychology and aging with computer science in understanding product comparison choices

Our results also show that users of product comparison tables frequently make suboptimal decisions. Even for medium-difficulty product choices, the mean frequency of optimal choices ranges from 61% to 70% (M = 70% for younger adults; M = 61% for middle-aged group; M = 62% for older adults). The difficulty of product comparison has the strongest impact on accuracy. In research in informatics that considered the objective quality of product choices, the choice of non-dominated alternatives was frequently used as a method to evaluate correctness [7, 10, 25]. However, this may be misleading, as choosing a non-dominated alternative is much easier than choosing among several non-dominated alternatives (making a compensatory choice).

The results of our experiments clearly prove the existence of individual, age-related differences in the correctness of product comparisons in e-commerce systems. The difference in the average correctness between younger and older adults can be as high as 19% (for the most-difficult tasks, when the average correctness of older consumers drops to 33%, while the correctness for younger consumers is 52%, and for middle-aged consumers 44%). This finding has important implications for informatics research, indicating the need for further decision support and UX design for product comparisons. Popular product comparison tables are clearly not sufficient for many users, especially older users. The understanding of data in a matrix form, using numerical values, and the subsequent requirement of computing a sum of products to use a strategy that can deal with compensatory choices seems to require cognitive abilities that not all users have. New, simpler and graphical forms of data presentation may be a solution to this problem.

### Limitations

Despite our best efforts, our study has some limitations. First, the experiment was conducted online, which means that older users who participated in the study must have IT competences and be sufficiently cognitively fit. However, the same applies for older users of e-commerce systems. Therefore, our results may not apply to all elderly individuals, but they should apply to older e-commerce users. The experimental design was realistic, but we did not use a real e-commerce store. We used data from real products and a realistic product comparison table. Following UX guidelines, we limited the number of attributes to 6 and alternatives to 3. This means that our results should not be affected by bad UX design. To determine the correct choices in the product comparison tasks, we told the users to use common preferences in the form of the importance ordering (weighting) of attributes.

We are aware that the proposed mediational moderation model, although original, is still preliminary, and elucidating the psychological mechanisms behind the role of VWM as a mediator (between age and decision accuracy) and intellectual helplessness as a moderator requires further experimental work (e.g., experimental manipulation of the intensity of a mediator and moderator, see [52].

We did not include price in the product attributes. We expect price to be the most important attribute overall, but its importance may vary strongly among individuals. Additionally, we expect prices to be strongly correlated with the other attributes. This would make it difficult to design product comparison tasks of controlled difficulty while accounting for attribute correlations. Instead of considering price directly as one of the attributes involved in product comparison, we assumed that e-commerce users compare products of similar prices. Each of the product comparisons in the experiments involves only high-end, or low-end products, reflected by the values of attributes of products in a comparison. There was no difference in our results for product comparisons of high-end or low-end products.

## Conclusions and future work

In this article, we have considered the impact of individual differences, particularly age and working memory, on the performance of users in product comparison choices. It links the disciplines of psychology (especially the domain of cognitive aging) and informatics. We have confirmed and extended findings from cognitive aging research on multi-attribute choice problems for e-commerce product comparison choices. It was predicted and confirmed that visual working memory was a reliable mediator of the relationship between age and decision accuracy. Importantly, there were significant correlations between visual working memory and decision accuracy for each age group. We also found intriguing findings that intellectual helplessness (of contracting an infectious disease) was a reliable moderator of the relationship between age and decision accuracy. The essence of this moderation effect is that the negative correlation between levels of intellectual helplessness and decision accuracy was significant in the young adult and middle-aged groups but not in the older adult group. It seems that older adults (but not younger and middle-aged people) are resistant to the deteriorating effects of intellectual helplessness (of contracting an infectious disease) on higher-order cognitive functions such as multi-attribute decision making.

Previous work in informatics has found that product comparison tables increase the likelihood that e-commerce users consider non-dominated products among their choices [7, 10, 25]. In our experiment, we have found a significant impact of the difficulty of comparison on its correctness. We define product choices as simple when any strategy can yield the right choice. However, this is equivalent to saying that among the compared products, there is a single product that dominates all others. In other words, when users choose more than one non-dominated product for comparison, they increase the difficulty of the resulting product choice problem. The conclusion from our results and from previous findings is that product comparison tables are a victim of their own success. By increasing the likelihood of choosing among non-dominated products, they increase the likelihood of making an incorrect choice. This is true for all users, regardless of age. Our experiment design follows the guidelines of correct interface design for product comparison tables [9]. Our findings show that these guidelines are not sufficient to remedy the inherent difficulties of product comparison. This finding calls for further research on improving the product comparison functionality, for example, by visual methods that do not require numerical processing.

On the other hand, age has a significant adverse effect on the ability of e-commerce users to make the right choice in product comparison. The fact that a particular part of the population

of e-commerce clients–older consumers–make suboptimal product choices more frequently creates a systematic bias in the data used for training recommender algorithms popular on e-commerce sites. This phenomenon has been called *self-induced bias* [32] and is a new type of data bias. State-of-the-art approaches toward removing biases that rely on increasing the frequency of data from the discriminated users would fail for this type of bias. New approaches are needed, as proposed by Pawłowska and colleagues [32] and studied using computer simulation. The new recommendation algorithms rely on the observation that working memory is an important mediator in the relationship between age and product choice optimality. Thus, algorithms can benefit from information about a user's working memory capacity, which can be obtained if the user performs a working memory test similar to the ones used in our experiment. However, more empirical research is necessary to confirm the effectiveness of the proposed new recommendation algorithms. New proposals of recommendation systems that are not biased against older e-commerce users can be the subject of future work.

Future work on the effects of cognitive limitations on product comparison may also benefit from the use of eye-tracking [29], in order to determine whether a user experiences cognitive fatigue, and to investigate what comparison strategy is used by experiment participants by following the focust of their sight on the products and features in the product comparison table.

## Supporting information

**S1 File.**
(DOCX)

## Author Contributions

**Conceptualization:** Klara Rydzewska, Radosław Nielek, Adam Wierzbicki, Grzegorz Sedek.

**Data curation:** Klara Rydzewska, Radosław Nielek, Adam Wierzbicki, Grzegorz Sedek.

**Formal analysis:** Klara Rydzewska, Radosław Nielek, Justyna Pawłowska, Adam Wierzbicki, Grzegorz Sedek.

**Funding acquisition:** Adam Wierzbicki, Grzegorz Sedek.

**Investigation:** Klara Rydzewska, Radosław Nielek, Justyna Pawłowska, Adam Wierzbicki, Grzegorz Sedek.

**Methodology:** Klara Rydzewska, Radosław Nielek, Justyna Pawłowska, Adam Wierzbicki, Grzegorz Sedek.

**Resources:** Klara Rydzewska.

**Software:** Radosław Nielek, Adam Wierzbicki.

**Supervision:** Adam Wierzbicki, Grzegorz Sedek.

**Validation:** Klara Rydzewska, Radosław Nielek, Adam Wierzbicki, Grzegorz Sedek.

**Visualization:** Radosław Nielek, Adam Wierzbicki.

**Writing – original draft:** Klara Rydzewska, Radosław Nielek, Adam Wierzbicki, Grzegorz Sedek.

**Writing – review & editing:** Klara Rydzewska, Radosław Nielek, Adam Wierzbicki, Grzegorz Sedek.

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
