## [Decision Letter · Decision Letter 0]

9 Feb 2024

PONE-D-23-38646Performance of Aging Consumers in an E-commerce Product Choice Task: The Role of Working Memory and Decision Strategies.PLOS ONE

Dear Dr. Nielek,

Thank you for submitting your manuscript to PLOS ONE. After careful consideration, we feel that it has merit but does not fully meet PLOS ONE’s publication criteria as it currently stands. Therefore, we invite you to submit a revised version of the manuscript that addresses the points raised during the review process. Please submit your revised manuscript by Mar 25 2024 11:59PM. If you will need more time than this to complete your revisions, please reply to this message or contact the journal office at plosone@plos.org. Please include the following items when submitting your revised manuscript:A rebuttal letter that responds to each point raised by the academic editor and reviewer(s). You should upload this letter as a separate file labeled 'Response to Reviewers'.A marked-up copy of your manuscript that highlights changes made to the original version. You should upload this as a separate file labeled 'Revised Manuscript with Track Changes'.An unmarked version of your revised paper without tracked changes. You should upload this as a separate file labeled 'Manuscript'.If applicable, we recommend that you deposit your laboratory protocols in protocols.io to enhance the reproducibility of your results. Protocols.io assigns your protocol its own identifier (DOI) so that it can be cited independently in the future. For instructions see: https://journals.plos.org/plosone/s/submission-guidelines#loc-laboratory-protocols. Additionally, PLOS ONE offers an option for publishing peer-reviewed Lab Protocol articles, which describe protocols hosted on protocols.io. Read more information on sharing protocols at https://plos.org/protocols?utm_medium=editorial-email&utm_source=authorletters&utm_campaign=protocols.

We look forward to receiving your revised manuscript.

Kind regards,

Sohaib Mustafa, Ph.D.

Academic Editor

PLOS ONE

Journal Requirements:

Reviewers' comments:

Reviewer's Responses to Questions

**Comments to the Author**

1. Is the manuscript technically sound, and do the data support the conclusions?

Reviewer #1: Yes

Reviewer #2: Yes

2. Has the statistical analysis been performed appropriately and rigorously? 

Reviewer #1: Yes

Reviewer #2: Yes

3. Have the authors made all data underlying the findings in their manuscript fully available?

Reviewer #1: Yes

Reviewer #2: Yes

4. Is the manuscript presented in an intelligible fashion and written in standard English?

Reviewer #1: Yes

Reviewer #2: Yes

5. Review Comments to the Author

Reviewer #1: It sheds light on the relationship between the age of users and their decision-making mechanisms on e-commerce platforms. In this context, It can be said that it contributes to the extant literature. Therefore, I accept this paper.

Reviewer #2: Dear Authors

Thank you for your submission.

Here are some remarks and recommendations regarding your manuscript titled “Performance of Aging Consumers in an E-commerce Product Choice Task: The Role of Working Memory and Decision Strategies”.

The proposed paper reflects experiments on purchase intent prediction for elderly customers. Some individual characteristics including visual working memory were shown as reliable mediators of the relationship between age and decision accuracy.

The results are based on a simulation and not real e-commerce data. Eye-tracking was not used in the study, which could have brought the results to the next level. All this should be thoroughly justified with view to updating the results in future papers based on real-world online shops and with more observation techniques.

The literature review is far too scarce for such a wide topic and lot of important related studies from recent years have been omitted by the authors. I would expect a much wider review of studies on purchase intent modeling based on user tracking including AI analytics such as fuzzy sets and neural networks. Perhaps also some papers where eye-tracking was used too. This should be revised and elaborated on.

Other than that, the paper is timely and interesting, in particular with regard to the recent pandemic, where online content started to play an increasingly important role. The article in general is technically sound, and the conceptual research model convincing with some more work in studying the state-of-the-art and justifying the methods.

I look forward to the improvements.

6. PLOS authors have the option to publish the peer review history of their article (what does this mean?). If published, this will include your full peer review and any attached files.

Reviewer #1: **Yes: **Mustafa Emre Civelek

Reviewer #2: No

---

## [Author Response · Author response to Decision Letter 0]

25 Mar 2024

We are writing this letter in response to the comments of reviewers of the article:

PONE-D-23-38646

Performance of Aging Consumers in an E-commerce Product Choice Task: The Role of Working Memory and Decision Strategies.

Submitted to PLOS ONE.

We would like to thank all reviewers for their careful consideration of the article, and for their insightful comments. We have revised the article taking into account all comments of all reviewers. In particular, the new version of the article includes the following changes:=

1. Addition of references about research for purchase intent modeling using behavioral and eye-tracking data

2. Formatting of paper according to journal guidelines, in particular

a. Headers

b. Removing figures from the main article, and leaving correct figure captions and references to figures according to the journal guidelines

c. References – adding DOI and PMID identifiers where possible

3. Checking figures’ compliance using Preflight Analysis and Conversion Engine (PACE) digital diagnostic tool. All submitted figures have been passed by PACE as compliant

Below please find detailed responses to reviewer’s comments:

Reviewer #1: It sheds light on the relationship between the age of users and their decision-making mechanisms on e-commerce platforms. In this context, It can be said that it contributes to the extant literature. Therefore, I accept this paper.

Thank you for your consideration and the positive review.

---

## [Decision Letter · Decision Letter 1]

1 May 2024

Performance of Aging Consumers in an E-commerce Product Choice Task: The Role of Working Memory and Decision Strategies.

PONE-D-23-38646R1

Dear Dr. Nielek,

We’re pleased to inform you that your manuscript has been judged scientifically suitable for publication and will be formally accepted for publication once it meets all outstanding technical requirements.

Kind regards,

Sohaib Mustafa, Ph.D.

Academic Editor

PLOS ONE

Additional Editor Comments (optional):

Reviewers' comments:

Reviewer's Responses to Questions

**Comments to the Author**

1. If the authors have adequately addressed your comments raised in a previous round of review and you feel that this manuscript is now acceptable for publication, you may indicate that here to bypass the “Comments to the Author” section, enter your conflict of interest statement in the “Confidential to Editor” section, and submit your "Accept" recommendation.

Reviewer #2: All comments have been addressed

2. Is the manuscript technically sound, and do the data support the conclusions?

Reviewer #2: Yes

3. Has the statistical analysis been performed appropriately and rigorously? 

Reviewer #2: Yes

4. Have the authors made all data underlying the findings in their manuscript fully available?

Reviewer #2: Yes

5. Is the manuscript presented in an intelligible fashion and written in standard English?

Reviewer #2: Yes

6. Review Comments to the Author

Reviewer #2: In my view, the authors have adequately addressed comments raised in a previous round of review. The paper is timely and interesting,

7. PLOS authors have the option to publish the peer review history of their article (what does this mean?). If published, this will include your full peer review and any attached files.

Reviewer #2: No

---

## [Editor Report · Acceptance letter]

12 Jun 2024

PONE-D-23-38646R1 

PLOS ONE

Dear Dr. Nielek, 

I'm pleased to inform you that your manuscript has been deemed suitable for publication in PLOS ONE. Congratulations! Your manuscript is now being handed over to our production team.

Kind regards, 

on behalf of

Dr. Sohaib Mustafa 

Academic Editor

PLOS ONE